# Exercise modifies glutamate and other metabolic biomarkers in cerebrospinal fluid from Gulf War Illness and Myalgic encephalomyelitis / Chronic Fatigue Syndrome

**James N. Baraniuk**[1]*, **Grant Kern**[1], **Vaishnavi Narayan**[1], **Amrita Cheema**[2]

1 Department of Medicine, Georgetown University, Washington, DC, United States of America,
2 Department of Oncology, Georgetown Lombardi Comprehensive Cancer Centre, Georgetown University, Washington, DC, United States of America

* baraniuj@georgetown.edu

**Data Availability Statement:** The minimal data sets are attached as Excel sheets in the Supporting Information files.

## Abstract

Myalgic encephalomyelitis / Chronic Fatigue Syndrome (ME/CFS) and Gulf War Illness (GWI) share many symptoms of fatigue, pain, and cognitive dysfunction that are not relieved by rest. Patterns of serum metabolites in ME/CFS and GWI are different from control groups and suggest potential dysfunction of energy and lipid metabolism. The metabolomics of cerebrospinal fluid was contrasted between ME/CFS, GWI and sedentary controls in 2 sets of subjects who had lumbar punctures after either (a) rest or (b) submaximal exercise stress tests. Postexercise GWI and control subjects were subdivided according to acquired transient postexertional postural tachycardia. Banked cerebrospinal fluid specimens were assayed using Biocrates AbsoluteIDQ® p180 kits for quantitative targeted metabolomics studies of amino acids, amines, acylcarnitines, sphingolipids, lysophospholipids, alkyl and ether phosphocholines. Glutamate was significantly higher in the subgroup of postexercise GWI subjects who did not develop postural tachycardia after exercise compared to nonexercise and other postexercise groups. The only difference between nonexercise groups was higher lysoPC a C28:0 in GWI than ME/CFS suggesting this biochemical or phospholipase activities may have potential as a biomarker to distinguish between the 2 diseases. Exercise effects were suggested by elevation of short chain acylcarnitine C5-OH (C3-DC-M) in postexercise controls compared to nonexercise ME/CFS. Limitations include small subgroup sample sizes and absence of postexercise ME/CFS specimens. Mechanisms of glutamate neuroexcitotoxicity may contribute to neuropathology and "neuroinflammation" in the GWI subset who did not develop postural tachycardia after exercise. Dysfunctional lipid metabolism may distinguish the predominantly female ME/CFS group from predominantly male GWI subjects.

**Funding:** JNB: W81XWH-15-1-0679, W81-XWH-09-1-0526, R21NS088138, R01NS085131. Department of Defense. https://cdmrp.army.mil/funding National Institute of Neurological Diseases and Stroke. https://www.ninds.nih.gov/ The funders had no role in study design, data collection and analysis, decision to publish, or preparation of the manuscript.

**Competing interests:** The authors have declared that no competing interests exist.

## Introduction

Myalgic encephalomyelitis / Chronic Fatigue Syndrome (ME/CFS) [1–3] and Gulf War Illness (GWI) [4–7] share many symptoms of fatigue, pain, and cognitive dysfunction that are not relieved by rest. Exerting more physical, cognitive, emotional or other effort than usual leads to symptom exacerbation that may be delayed 24 hr or more in onset, and that is not relieved by rest; this has been termed post-exertional malaise or exertional exhaustion. ME/CFS has a sporadic incidence and predominantly affects females (4:1), while GWI affects 25% to 32% of the cohort of predominantly male personnel exposed to conditions in the Persian Gulf in 1990–1991. The overlap in subjective symptoms and criteria highlights the need for objective biomarkers to advance understanding of pathophysiological mechanisms, improve diagnostics that distinguish between the two diseases, and develop specific treatment strategies.

Metabolomics alterations have been found in the brain [8] and peripheral blood from ME/CFS [9–19] and GWI subjects [20, 21]. The patterns of change in amino acids, glycolysis, fatty acids, and other mediators implicate mitochondrial dysfunction that may affect glyceraldehyde-3-phosphate dehydrogenase (GAPDH), pyruvate dehydrogenase and its inhibitors, transport of metabolites across the mitochondrial membranes, interruption of electron transport chain complexes, or peroxisomal metabolism of very long chain polyunsaturated fatty acids. The molecular details responsible for dysfunction remain to be defined for each disease.

Metabolomic profiles in cerebrospinal fluid were assessed as part of a larger analysis of proteomic [22], neuropeptide [23–25], miRNA [26] and other biomarker alterations in ME/CFS and GWI compared to sedentary control (SC) subjects.

Lumbar punctures were performed in 2 settings in different groups of participants. First, "nonexercise" subjects rested overnight before history and physical, phlebotomy and lumbar puncture (nonexercise groups abbreviated as *sc0*, *cfs0* and *gwi0*) [26]. Second, a model of exertional exhaustion was developed by having subjects perform submaximal bicycle exercise on 2 consecutive days with functional magnetic resonance imaging (fMRI) studies performed at baseline and after the stress tests [27–33]. Lumbar puncture was performed after the post-exercise MRI in sedentary control (**SC**) and GWI subjects.

GWI subjects were subdivided into 2 groups based on transient postural tachycardia that developed after exercise. One third of participants developed postural tachycardia after exercise because their heart rates rose by ≥30 beats per minute after standing up [27, 34, 35]. They were termed the Stress Test Activated Reversible Tachycardia (START) phenotype. This response was significantly different from Postural Orthostatic Tachycardia Syndrome (POTS) [36, 37] because START had normal elevations of 10 to 15 beats per minute before exercise, then transient postural tachycardia for 1 to 3 days after exercise.

The other two thirds of GWI subjects had normal postural heart rate elevations at all times, and significant patterns of fMRI changes following exercise that were reminiscent of phantom limb and other pain syndromes [38, 39] and significantly different from START and controls; this GWI subset was called the Stress Test Originated Phantom Perception (STOPP) phenotype. The post-exercise groups were abbreviated in capitals as **SC**, **START** and **STOPP** for comparison to the nonexercise groups.

The START phenomenon was also induced by exercise in ME/CFS and control groups [34, 35]. However, **START** versus **STOPP** status was not associated with postexercise changes in fMRI in control or ME/CFS [31–33].

Cerebrospinal fluid metabolomics data were assessed for differences between nonexercise (*sc0*, *cfs0* and *gwi0*) and postexercise (**SC**, GWI **START** and **STOPP**) groups, and between post-exercise and appropriate nonexercise controls (**SC** vs. *sc0*, **START** vs. *gwi0*, **STOPP** vs. *gwi0*). Nonexercise subjects could not be assigned into **START** or **STOPP** phenotypes because

exercise is necessary to induce the defining postural tachycardia. No post-exercise ME/CFS cerebrospinal fluid specimens were available for this pilot investigation.

## Methods

### Subjects

Protocols were approved by the Georgetown University Institutional Review Board (2006–481, 2009–229, 2013–0943, 2015–0579) and USAMRMC Human Research Protection Office (HRPO #A-15547, A-18479), and listed in clinicaltrials.gov (NCT01291758, NCT00810225, NCT03342482).

Healthy and Gulf War Illness (GWI) veterans from the 1990–1991 Persian Gulf War, ME/CFS, and healthy, non-military control subjects were recruited from websites, word of mouth, fliers, newspaper and online advertisements, and personal contacts in clinics and support groups. Interested participants responded via telephone or email. After obtaining verbal consent, each volunteer had an initial telephone screening with a clinical research associate who read a scripted outline of the study to assess inclusion and exclusion criteria. Candidates were screened for military service in the Gulf War Theater; Center for Disease Control (CDC) criteria for Chronic Multisymptom Illness (CMI) [5] and Chronic Fatigue Syndrome (CFS) [1]; Kansas criteria for GWI [6]; current medications; chronic medical and psychiatric illnesses [40–42]. Eligible subjects came to the Georgetown Howard Universities Clinical and Translational Science Clinical Research Unit where diagnosis and study inclusion were confirmed by history and physical examination.

GWI was defined by moderate to severe scores for (a) ≥2 out of 3 CMI criteria (fatigue, mood-cognition, and musculoskeletal pain) [5] plus (b) ≥3 out of 6 symptom domains of the Kansas criteria (fatigue/sleep, pain, neurological/cognitive/mood, gastrointestinal, respiratory, skin) [6, 7, 43].

ME/CFS was defined using 1994 CDC "Fukuda" criteria [1] plus Canadian Consensus Criteria [2, 3]. The CDC criteria require disabling fatigue lasting more than 6 months that cannot be explained by exclusionary medical or psychiatric diagnoses [8–10] plus at least 4 of 8 ancillary symptoms: short term memory of concentration problems, sore throat, sore lymph nodes, myalgia, arthralgia, headache, sleep disturbance, and post-exertional malaise (exertional exhaustion). The Carruthers Canadian Consensus Criteria emphasized fatigue, postexertional malaise, sleep, pain, cognition, and added an array of flu-like, autonomic and interoceptive symptoms [2, 3]. Subjects were not assessed for 2015 Institute of Medicine criteria for Systemic Exertion Intolerance Disease (SEID) with fatigue, postexertional malaise, sleep and either cognitive or orthostatic problems [44].

Sedentary controls in the nonexercise (*sc0*) and postexercise (*SC*) groups had a sedentary lifestyle with less than 40 min of aerobic exercise per week and did not meet ME/CFS, CMI or Kansas criteria.

Exclusion criteria included substance abuse, hospitalization for a psychiatric disorder in the past 5 years, or a chronic medical or psychiatric condition not related to service in the Persian Gulf [40–42].

Two cohorts were studied. The first day of the 2 protocols was considered an adjustment period, and included the patient's history and physical, blood work, and baseline studies. The first "nonexercise" cohort rested before having lumbar puncture, and were termed *sc0*, *cfs0* and *gwi0* groups. They did not have exercise.

The second "postexercise" cohort had magnetic resonance imaging (MRI), submaximal bicycle exercise stress testing, and serial assessments of postural tachycardia. They rested overnight, then had their second identical stress test, MRI and "postexercise" lumbar puncture.

Subjects cycled at 70% of predicted heart rate (pHR = 220-Age) for 25 min then increased to 85% pHR. Exercise-induced postural tachycardia defined two subgroups of GWI veterans [27, 34]. Exercise was required to define **START** vs. **STOPP** phenotypes. The post-exercise sedentary control group was termed **SC**.

## Lumbar puncture

Identical methods were used in all subjects. Cerebrospinal fluid (20 ml) was drawn from the L1-L2 interspace using Gertie-Marx needles [45, 46] in the prone position under fluoroscopic guidance by interventional radiologists. Specimens were immediately placed on ice, centrifuged at 4˚C, specimens sent for routine laboratory studies, and aliquoted and stored at -80˚C within 1 hr. After lumbar puncture subjects were informed strictly to rest in comfortable positions and to avoid straining actions and Valsalva maneuvers such as lifting luggage for 24 hr before discharge.

## Metabolomics

Biocrates AbsoluteIDQ® p180 kit was employed according to manufacturer's instructions to detect amino acids (n = 21), biogenic amines (n = 21), total hexoses, acylcarnitines (n = 40), lysophosphatidylcholines (n = 14), phosphatidylcholines (n = 76), and sphingolipids (n = 15) [47–49]. Frozen (-80˚C) deidentified cerebrospinal fluid specimens were analyzed in blinded fashion by adding specimens (35 μL) and internal standard solution (10 μL) to wells of each 96-well extraction plate. Low, medium and high quality control samples, blank, zero sample, and calibration standards were added to appropriate wells. The plate was dried under a gentle stream of nitrogen. Samples were derivatized with phenyl isothiocyanate to detect amino acids and biogenic amines, then extracted with 5mM ammonium acetate in methanol before being diluted 15:1 with 40% methanol in water for the UPLC-MS/MS analysis of amino acids and amines, and 20:1 with proprietary kit running solvent for flow injection analysis (FIA) mass spectrometry of acylcarnitines, sphingolipids and glycerophospholipids.

Chromatographic separation of amino acids and biogenic amines and their stable-isotope labeled internal standards was performed on a ACQUITY UPLC System (Waters Corporation) using a ACQUITY 2.1 mm x 50 mm 1.7 μm BEH C18 column fitted with a ACQUITY BEH C18 1.7 μm VanGuard guard column, and quantified by calibration curve using a linear and quadratic regression with 1/x weighting [50, 51]. Eluates from UPLC and FIA were introduced directly into positive electrospray ionization device and Xevo TQ-S mass spectrometer (Waters Corporation) operating in the Multiple Reaction Monitoring (MRM) mode. MRM transitions (compound-specific precursor to product ion transitions) for each analyte and its internal standard were collected over the appropriate retention time using tune files and acquisition methods provided in the AbsoluteIDQ® p180 Standard Operating Procedures [47]. The UPLC data were imported into TargetLynx (Waters Corporation) for peak integration, calibration and concentration calculations (μM). Peaks were quantified using FIA and the processed TargetLynx data and Biocrates' Workflow Manager MetIDQ™ system. Data were analyzed without imputation for levels at or below the lower limits of detection. Lipids were named using LIPID MAPS Structure Database [50–53] to identify potential isobaric (± 0.5 Da) and isometric lipid species.

## Statistical and bioinformatic analysis

The primary variable of interest for general linear regression was the group identifier in order to find significant differences between **sc0**, **cfs0**, **gwi0**, **SC**, **START** and **STOPP** groups. Gender, body mass index (BMI), and age have been associated with specific metabolites in previous

studies [54–56] and these variables and their cross-products were incorporated into the models. Group differences in cerebrospinal fluid metabolite concentrations were assessed by ANOVA with Tukey Honest Significant Difference tests to correct for multiple comparisons (SPSS24 software). Effect sizes were estimated with Hedges' g [57]. Pearson correlations, principal component analysis (PCA), and t-distributed stochastic neighbor embedding (t-SNE) analysis [58] (R Studio Rtsne package) [59] were performed to check for collinearity, possible technical artifacts and any other unusual patterns.

The primary raw data are provided in S1 Raw data.

## Results

Study candidates (n = 442) were extensively screened (Table 1). The 116 inpatient study participants were predominantly white (n = 103) and non-Hispanic (n = 109) (Table 2) [60]. Five of the seven post-exercise *SC* subjects were deployed Persian Gulf War veterans. The high symptom load in ME/CFS and GWI was confirmed by the "floor" effects of *sc0* and *SC* groups compared to the elevated "ceiling" effects for symptom scores and disabling (lower) quality of life scores in *cfs0*, *gwi0*, *START* and *STOPP* (Table 2).

Pearson correlations were assessed between the concentrations of all 180 metabolites in each of the 6 groups (Fig 1, S1 to S6 Tables). Phospholipids from all classes had the highest Pearson correlations in *SC* and *sc0*. Amino acids had relatively high correlations in *START*, *sc0*, *SC* and *gwi0* (p > 0.8). *SC* had negative correlations between several amino acids and carnitines (p < -0.5). Only *START* had positive correlations between amino acids and phospholipids. Principal component analysis did not reveal any noteworthy relationships between metabolites, disease, groups, exercise, gender or batches.

t-SNE of the Pearson correlation coefficients between analytes (Fig 1) generated 4 clusters of groups (Fig 2). Nonexercise females and males were separated which was consistent with differences in relative activities of specific metabolic pathways between men and women [54–

**Table 1. CONSORT information [60].**

| Study protocol | Nonexercise lumbar puncture | Exercise protocol followed by lumbar puncture |
|---|---|---|
| Subjects recruited | Sedentary control (*sc0*) | Sedentary control (*SC*) |
| | CFS (*cfs0*) | GWI *START* phenotype was defined by postural tachycardia after exercise. *STOPP* had no postural tachycardia. |
| | GWI (*gwi0*) | |
| Contacted | 271 | 171 |
| Screened | 123 | 96 |
| Medical exclusions | 13 | 16 |
| Declined to participate | 24 | 51 |
| Lumbar puncture | 86 (57 female) | 30 (4 female) |
| Groups and N (N females) | | |
| *sc0* | 21 (10 female) | 0 |
| *cfs0* | 45 (36 female) | 0 |
| *gwi0* | 20 (11 female) | 0 |
| *SC* | 0 | 7 (0 female) |
| *START* phenotype of GWI | 0 | 7 (1 female) |
| *STOPP* phenotype of GWI | 0 | 16 (3 female) |
| Total | 116 (61 female) | |

**Table 2. Demographics.**

| | Nonexercise | | | Post-exercise | | |
|---|---|---|---|---|---|---|
| | *sc0* | *cfs0* | *gwi0* | *SC* | *START* | *STOPP* |
| n (female) | 21 (10) | 45 (36) | 20 (11) | 7 (0) | 7 (1) | 16 (3) |
| Age | 42.3±5.6 | 45.6±3.1 | 48.9±4.4 | 50.1±5.9 | 45.1±7.6 | 44.7±3.2 |
| BMI | 27.0±2.3 | 28.8±2.1 | 30.8±2.3 | 30.0±2.1 | 29.0±4.0 | 29.7±2.7 |
| CFSQ | | | | | | |
| fatigue | 1.2±0.6 | 3.7±0.2† | 3.5±0.2† | 1.1±0.5 | 3.6±0.4† | 3.5±0.3† |
| memory | 1.0±0.5 | 3.1±0.2† | 2.8±0.4† | 1.0±0.9 | 3.0±0.7† | 2.8±0.4† |
| sore throat | 0.4±0.3 | 1.4±0.4* | 1.4±0.5 | 0.1±0.3 | 2.4±0.6† | 1.5±0.6* |
| sore nodes | 0.2±0.3 | 1.3±0.4 | 1.6±0.7† | 0.0±0.0 | 2.0±1.0† | 1.2±0.5 |
| muscle pain | 1.2±0.6 | 2.9±0.4† | 3.2±0.5† | 0.9±0.5 | 3.3±0.4† | 3.1±0.5† |
| joint pain | 0.8±0.4 | 2.6±0.4† | 2.8±0.6† | 1.1±0.7 | 3.0±1.0† | 3.3±0.4† |
| headaches | 0.7±0.5 | 2.5±0.4† | 2.5±0.5† | 1.1±1.1 | 3.1±0.9† | 2.7±0.6† |
| sleep | 1.4±0.7 | 3.5±0.3† | 3.5±0.3† | 1.9±0.9 | 3.4±0.4† | 3.6±0.5† |
| exertional exhaustion | 1.2±0.7 | 3.4±0.3† | 3.2±0.6† | 0.4±0.6 | 3.6±0.4† | 3.2±0.5† |
| Sum of 8 CFS ancillary criteria scores | 7.0±3.1 | 20.7±1.3† | 20.9±2.4† | 6.6±3.1 | 24.1±3.1† | 21.5±2.8† |
| SF-36 | | | | | | |
| Physical functioning | 85.5±10.1 | 41.5±6.8† | 42.2±12.6† | 80.0±21.3 | 43.6±22.6† | 46.1±12.5† |
| Role physical | 65.5±19.6 | 6.3±5.8† | 9.7±9.0† | 89.3±14.6 | 0.0±0.0† | 19.6±17.9† |
| Bodily pain | 78.2±13.2 | 32.4±5.9† | 37.6±12.3† | 70.9±16.2 | 19.7±12.7† | 30.4±10.8† |
| General health | 71.4±7.8 | 34.5±6.0† | 28.1±7.8† | 75.3±8.8 | 13.9±8.7†¶ | 31.3±11.5† |
| Vitality | 57.4±12.2 | 12.1±3.5† | 22.2±6.2† | 65.7±11.2 | 15.0±8.8† | 15.0±7.7† |
| Social functioning | 82.1±12.1 | 25.0±6.7† | 29.9±11.2† | 85.7±12.5 | 16.1±12.8† | 33.0±11.1† |
| Role emotional | 82.5±14.7 | 58.3±14.0 | 25.9±18.0† | 90.5±18.7 | 4.8±9.3†¶ | 35.7±23.2† |
| Mental health | 77.9±4.8 | 64.2±5.4 | 51.1±10.9† | 74.9±12.4 | 41.1±15.9†¶ | 55.4±9.1† |
| IgG/Albumin | 0.124 ±0.019 | 0.113 ±0.012 | 0.124 ±0.013 | 0.122 ±0.018 | 0.112 ±0.022 | 0.114 ±0.019 |

*p<0.05 and

†p<0.01 vs. sc0 and SC, and

¶p<0.05 vs. cfs0 by Tukey Honest Significant Difference.

mean ± 95% CI.

56] that were not influenced by ME/CFS or GWI status. The male-dominated **SC**, **START** and **STOPP** data were separated from nonexercise males suggesting an exercise effect. The postexercise **SC** group was distinct from the predominantly male GWI **START** and **STOPP** groups suggesting an interaction of exercise and GWI status. Gender, exercise and disease effects were factored into general linear modeling (GLM).

## Glutamate

Glutamate was significantly elevated in the GWI **STOPP** subgroup compared to nonexercise **sc0**, **cfs0**, **gwi0**, and postexercise **START** using univariate GLM to correct for subject group, age, gender and BMI (adjusted p = 0.001) followed by ANOVA (p = 0.0020) and Tukey Honest Significant Difference between groups (p≤0.024) (Fig 3). There was no indication of any differences between nonexercise groups, or for a bimodal distribution of glutamate in nonexercise GWI subjects that could have predicted **START** versus **STOPP** status. Elevation of glutamate in the postexercise **STOPP** group suggested effects related to the exertion, GWI as an underlying pathology, and some aspect of the maintenance of the normal postural tachycardia response following exercise. **START** was equivalent to the nonexercise groups indicating

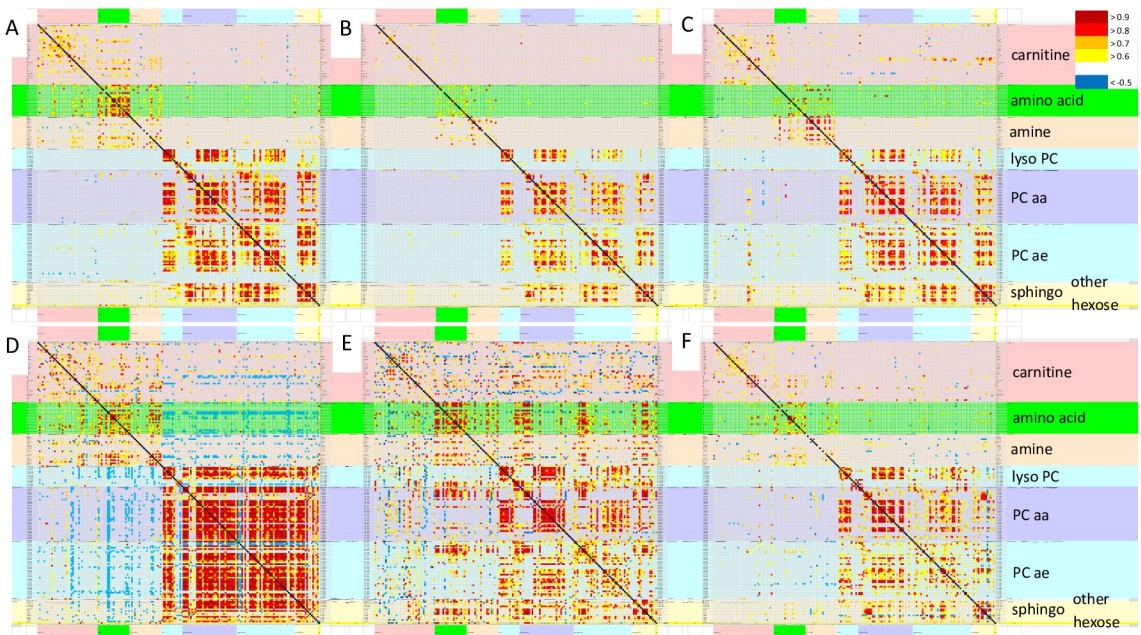

**Fig 1. Pearson correlations between all 180 analytes.** Pearson correlation matrices were depicted for *sc0* (A), *cfs0* (B), *gwi0* (C), *SC* (D), *START* (E) and *STOPP* (F). The line of identify was black. Metabolite classes and legend were color coded (right). Metabolites were arranged in alphabetical order for each class. Each panel is shown in more detail in the S1 to S6 Tables.

that the elevation of glutamate in *STOPP* was not directly related to the development of post-exertional postural tachycardia. One individual in each of the *cfs0*, *gwi0* and *STOPP* groups had glutamate levels that were more than the 95th percentiles ($2\sigma$) for each group. Removing

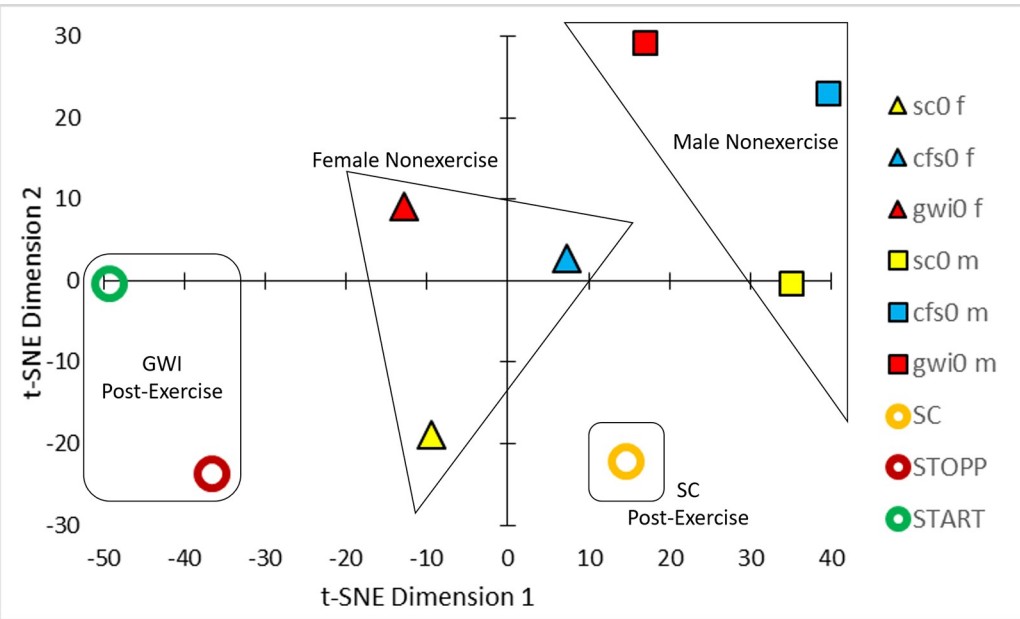

**Fig 2. t-SNE.** Analysis of the Pearson correlation coefficients between analyte concentrations from the 6 groups (Fig 1) led to clusters of nonexercise females (triangles), nonexercise males (squares), postexercise controls (orange circle), and postexercise *START* (green circle) and *STOPP* (red circle) subgroups of GWI subjects.

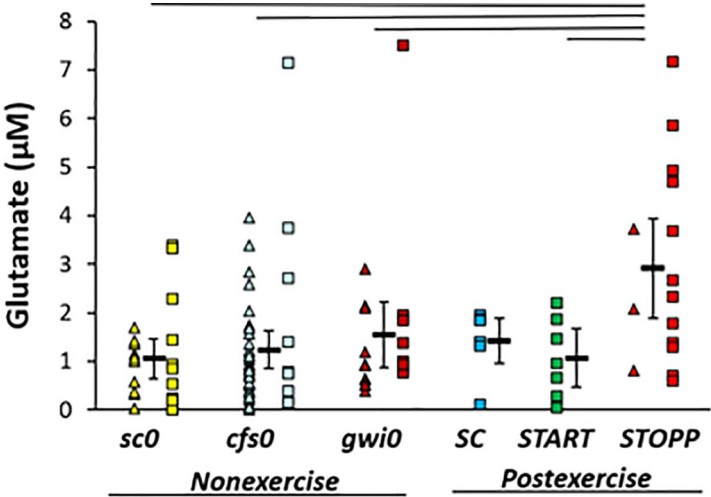

**Fig 3. Glutamate in cerebrospinal fluid.** Glutamate was significantly higher in *STOPP* (bright red) than nonexercise *sc0* (yellow), *cfs0* (light blue), *gwi0* (dark red), and postexercise *START* (green). Females (triangles), males (squares) and group mean ± 95% CI (black bar) were shown for each group. *SC* had no females (no triangles). The lines above the bars indicate significant Tukey's tests (p ≤ 0.024). Hedges' g was 1.3 for *STOPP* versus all other groups.

these subjects did not substantially alter the statistical relationships, and so their data were retained. Hedges' g was 1.3 for the difference between *STOPP* and all other data.

*SC > sc0.* C5-OH (C3-DC-M, 3-hydroxyisovaleryl-carnitine, 2-methyl-3-hydroxybutyryl-carnitine) was significantly higher in SC than sc0 by GLM (p corrected = 0.048; Tukey p = 0.009) indicating an exercise effect in sedentary control subjects (Fig 4). Hedges' g was 1.36 for *SC* compared to all other subjects and 1.65 versus *sc0*.

*gwi0 > cfs0.* LysoPC a C28:0 was significantly higher in *gwi0* than *cfs0* by ANOVA and Tukey tests (p = 0.036) (Fig 5). In post hoc analysis, *gwi0* was significantly higher than the combination of *START* and *STOPP* (all postexercise GWI subjects, 2-tailed, unpaired t-test p = 0.036). Hedges' g was 0.8 for *gwi0* compared to all other subjects.

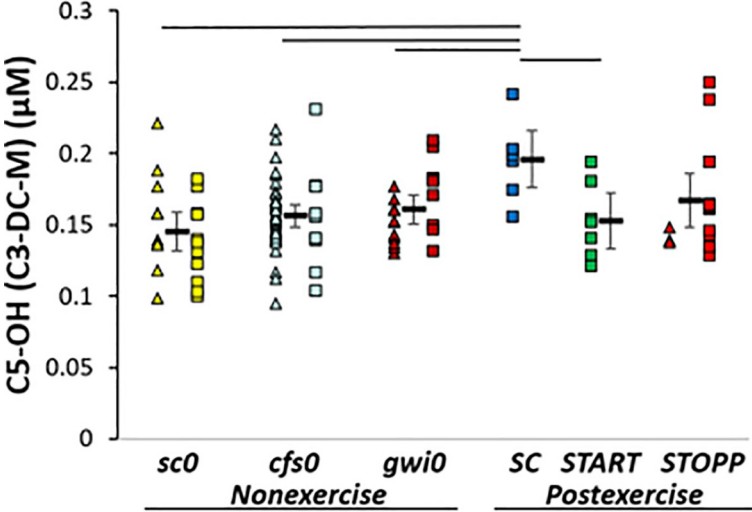

**Fig 4. C5-OH.** C5-OH (C3-DC-M) was higher in *SC* than other groups by GLM, ANOVA and Tukey tests (p < 0.05).

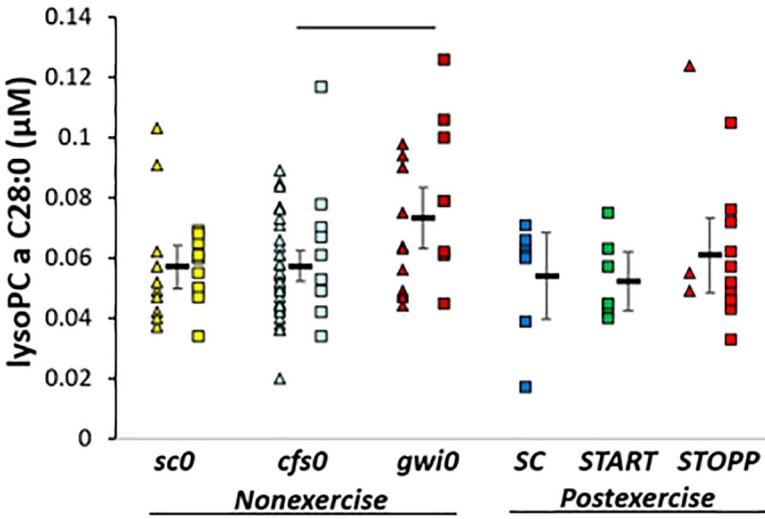

**Fig 5. LysoPC a C28:0.** Levels were higher in nonexercise *gwi0* than *cfs0*. The bar indicates significance by Tukey test (p = 0.036).

## Discussion

Glutamate was significantly increased in the GWI *STOPP* compared to all nonexercise and the postexercise *START* groups. This result suggests that mechanisms of increased neural excitotoxicity are activated by the exercise stressor in this majority phenotype of GWI. *START* did not have this change indicating exercise alone was not responsible. Instead, elevated glutamate may have been the result of an interaction of exercise with as yet unknown pathological mechanisms active in the *STOPP* phenotype. Glutaminergic excitotoxicity is highly relevant for interpreting the benefits of a low glutamate diet in GWI and irritable bowel syndrome [61–63].

Cerebrospinal fluid was different from serum where ME/CFS have reduced glutamine, ornithine and glucogenic amino acids [9]. This may be due to the differences in glutamine shuttle, energy metabolism and high oxygen requirements of neurons and glial cells compared to peripheral tissues [64] and disruption of the excitatory: inhibitory balance between glutamate and gamma-aminobutyric acid (GABA) in neural processing [65]. There were no compensatory changes in antioxidant peptides such as carnosine (β-alanyl-histidine) and anserine (β-alanyl-N-methylhistidine). N-acetyl-aspartic acid (NAA) was not altered, but is relevant as a source of acetyl-CoA for myelin lipid synthesis in oligodendrocytes, ATP generation in neuronal mitochondria, and as a precursor for the brain neurotransmitter N-acetylaspartylglutamate that is also an alternative source of glutamate [66]. N-acetylaspartylglutamate is a neuron osmolyte and transmitter for axon-glial signaling [67]. GABA, homocarnosine (GABA-histidine) and N-acetylaspartylglutamate were not measured by the Biocrates panel indicating a need to expand the scope of analytes screened using targeted metabolomics of cerebrospinal fluid.

Acylcarnitine C5-OH (C3-DC-M) was higher in postexercise *SC* than other groups. This suggested (i) no differences between nonexercise groups, (ii) effects of exercise (*SC*>*sc0*) and (iii) differences between postexercise *SC* and GWI groups.

Steady state nonexercise acylcarnitine levels were equivalent in the 3 nonexercise groups (*sc0, cfs0, gwi0*). Serum levels of C8:1, C12DC, C14, C16:1, C18, C18:1 (oleyl-L-carnitine), C18:2 (linoleyl-L-carnitine) and C18:1-OH acylcarnitines were significantly lower in ME/CFS

(n = 44) than controls (n = 49, p < 0.0001) [68, 69]. Levels were correlated with clinical symptoms [70]. Japanese and Swedish ME/CFS groups had significant reductions after correcting for diet and nationality [71]. Very long-chain acylcarnitines were decreased in veterans with GWI [20, 21]. These data suggest the hypothesis that reduced carnitine palmitoyltransferase-I (CPT-I) activity and mitochondrial fatty acid transport may contribute to the pathogenesis of ME/CFS and GWI, and that appropriate nutrient supplementation may correct the imbalances. However, this hypothesis is controversial as other studies found no differences for total carnitine, free carnitine, acylcarnitine and carnitine esters in 25 female ME/CFS and 25 healthy matched neighbourhood controls [72] nor in plasma and timed urine samples from ME/CFS (n = 31), healthy controls (n = 31), depression (n = 15) and rheumatoid arthritis (n = 22) [73]. Elevated serum and urine C5-OH are indicators of marginal biotin deficiency in humans [74] but this is an unlikely explanation in the postexercise control subjects. Cerebrospinal fluid / plasma ratios for L-carnitine and C3-carnitine, but not C5, were significantly elevated in Alzheimer disease compared to controls [75]. Low intramuscular levels of short-chain acylcarnitines have been found in older females compared to males [76]. Low levels were correlated with reduced physical performance and deficient expression of genes involved in mitochondrial energy production. It is not clear if serum or urine levels correlate with cerebrospinal fluid, if C5-OH, other short chain, moderate and long chain acylcarnitines are altered in parallel, or if differences between radioenzymatic, liquid chromatography, mass spectrometry and nuclear magnetic resonance methods may account for some of the diversity of outcomes.

The exercise protocol of 25 minutes of submaximal bicycling at 70% predicted maximum heart rate was comparable to studies with 30 min duration at 40% to 45% of $VO_{2MAX}$ that rapidly activate muscular work, lipolysis, robust fatty acid oxidation [77], and elevate short chain acylcarnitines in plasma [78] while minimizing the contribution from liver β-oxidation [79]. Higher C5-OH in postexercise *SC* than *sc0* would be consistent with this metabolic exercise effect. However, plasma metabolites were not measured and we have not established whether the source of cerebrospinal fluid acylcarnitines was flux across blood-brain barriers from plasma or production by central nervous system neurons or glial cells. In contrast, there were no differences in carnitines between nonexercise *gwi0* and postexercise GWI subjects suggesting that exercise did not increase carnitine production by fatty acid oxidation or other pathways in that disease.

The higher C5-OH levels in *SC* than postexercise GWI supports the contention that GWI subjects had inefficient fatty acid oxidation that did not respond appropriately to the exercise stimulus. Another alternative may be disordered metabolism of leucine or other anaplerotic pathways [19] in muscle, brain or other organs during and after exercise. An unlikely explanation may be differences in timing of meals before lumbar puncture in each group. Caution is needed to not overinterpret the C5-OH results based on the finding that plasma amino acid and acylcarnitine levels can change dynamically within minutes [80, 81].

Only one other lipid was significant. lysoPC a C28:0 was higher in nonexercise *gwi0* than *cfs0*. Further study is needed to determine if this lipid may be a biomarker to differentiate between mechanisms of GWI and ME/CFS. Unlike plasma [20], the sums of concentrations for all carnitine, lysophospholipid, phosphatidylcholine and sphingomyelin lipid families were not significantly different between GWI, ME/CFS, control, or exercise groups.

There are significant limitations to interpretation of this pilot study. The nonexercise and postexercise sets of cerebrospinal fluid were collected at different times with different lots of lumbar puncture kits. Lumbar punctures were performed between 1 and 4 pm, so the timing of lunch and the postprandial time periods may differ between postexercise and nonexercise subjects leading to differences in circulating nutrients, gut metabolites, hormones, and fluids. Different groups of nonexercise and postexercise subjects were tested in parallel because time

course studies of individuals who had lumbar punctures before and immediately after exercise would have had high risks for postexertional Valsalva-related spinal headaches. Sample collection was standardized, but prolonged storage may have allowed alteration of analyte concentrations and stoichiometry even though some compounds such as carnitines are reported to be stable indefinitely when stored at -80˚C [82]. Cerebrospinal fluid originates as a plasma exudate through choroid plexus epithelial cells with alteration of metabolite mixes by neurons and glia. The contributions and mechanisms of release from each cellular source cannot be directly traced in these subjects.

The Biocrates kit cannot differentiate lipids attached to the glycerol backbone in sn-1 versus sn-2 positions or positions of acyl versus alkyl bonds. The standards provide the total number of carbon atoms and double bonds but cannot distinguish between different isomers (same molecular weight but different chemical structures), isobars (same nominal mass using most abundant isotopes, but with different exact masses), relative lengths of chains at either location, or the exact positions of double bonds. Sphingomyelins are named based on the adduct to the amide bond and its numbers of carbon atoms, double bonds and presence or absence of a hydroxyl group with the assumption that the backbone was sphingosine d18:0. Many other side chains were possible at this position, and would have to be identified using untargeted metabolomics strategies. Phosphocholines and lysophosphocholines with alkyl and ether linkages were investigated, but the positions of the ether links were not specified. Other lipid classes identified from the head groups such as ethanolamine and inositol phosphatides, sulfotides, and glycophospholipids were not characterized. Quantification of these lipids will require future studies with orthogonal detection and characterization (e.g. nuclear magnetic resonance), larger test sets, and spiking with predefined pure compounds. The scope of chemicals will also need to be expanded to include oxidized and nitrosylated markers of reactive free radicals such as 4-hydroxynonenal (4-HNE) and malondialdehyde (MDA), peroxidation of polyunsaturated fatty acids (PUFAs) such as linoleic, arachidonic, and docosahexaenoic acids that contain the highly susceptible (1Z, 4Z) pentadiene moiety [83].

The ***STOPP–START*** dichotomy in GWI was associated with differences in postexercise cerebrospinal fluid glutamate (Fig 1), miRNA [26] and differential activation of cerebellar and other brain regions by fMRI [31–33]. This dichotomy does not appear to be as influential in ME/CFS or control subsets. Additional work is required to deconstruct these processes into their component parts and to develop new mechanistic hypotheses to explain the exercise-induced consequences.

## Conclusion

Glutamate was elevated in the GWI ***STOPP*** phenotype suggesting the hypothesis that these subjects have neuroexcitotoxicity as a component of postexertional malaise and exertional exhaustion. Future studies are required to determine if cerebrospinal fluid levels of glutamate become elevated in the postexercise STOPP subset of ME/CFS or if this change is limited to GWI. Exertional stressors may exacerbate dysfunctional brainstem autonomic control systems, astrocytes, amino acid metabolism and mitochondrial energy production. Hedges' g suggests the elevated glutamate measurement should be reproducible if the testing paradigm is repeated or the samples assayed with independent methods. Lyso PC a C28:0 may differentiate ME/CFS from GWI. C5-OH was elevated in the postexercise control group and may be a marker of exercise effects. These findings are consistent with hypotheses of dysfunctional amino acid and mitochondrial energy production, lipid and peroxisome metabolism with changes in carnitines, phospholipase enzyme activities, lysophospholipids and polyunsaturated fatty acids in ME/CFS [8–19, 68–73] and GWI [20, 21].

## Supporting information

**S1 Raw data.**
(XLSX)

**S1 Table.**
(XLSX)

**S2 Table.**
(XLSX)

**S3 Table.**
(XLSX)

**S4 Table.**
(XLSX)

**S5 Table.**
(XLSX)

**S6 Table.**
(XLSX)

## Author Contributions

**Conceptualization:** James N. Baraniuk, Amrita Cheema.

**Data curation:** James N. Baraniuk, Grant Kern, Vaishnavi Narayan, Amrita Cheema.

**Formal analysis:** James N. Baraniuk, Grant Kern, Vaishnavi Narayan, Amrita Cheema.

**Funding acquisition:** James N. Baraniuk.

**Investigation:** James N. Baraniuk, Amrita Cheema.

**Methodology:** James N. Baraniuk, Amrita Cheema.

**Project administration:** James N. Baraniuk.

**Resources:** James N. Baraniuk, Amrita Cheema.

**Software:** James N. Baraniuk, Amrita Cheema.

**Supervision:** James N. Baraniuk.

**Validation:** James N. Baraniuk, Amrita Cheema.

**Visualization:** James N. Baraniuk, Grant Kern, Vaishnavi Narayan.

**Writing – original draft:** James N. Baraniuk, Grant Kern, Vaishnavi Narayan, Amrita Cheema.

**Writing – review & editing:** James N. Baraniuk, Vaishnavi Narayan, Amrita Cheema.

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
