## [Decision Letter · Decision Letter 0]

20 Oct 2020

PONE-D-20-29578

Metabolic biomarkers in cerebrospinal fluid from Gulf War Illness and Myalgic Encephalomyelitis / Chronic Fatigue Syndrome

PLOS ONE

Dear Dr. Baraniuk,

Thank you for submitting your manuscript to PLOS ONE. After careful consideration, we feel that it has merit but does not fully meet PLOS ONE’s publication criteria as it currently stands. Therefore, we invite you to submit a revised version of the manuscript that addresses the points raised during the review process. Specifically, please improve the flow of the manuscript and provide additional descriptions for the points raised by the reviewers.

We look forward to receiving your revised manuscript.

Kind regards,

Pankaj K Singh, Ph.D.

Academic Editor

PLOS ONE

Journal Requirements:

2)  In your Data Availability statement, you have not specified where the minimal data set underlying the results described in your manuscript can be found. PLOS defines a study's minimal data set as the underlying data used to reach the conclusions drawn in the manuscript and any additional data required to replicate the reported study findings in their entirety. All PLOS journals require that the minimal data set be made fully available. For more information about our data policy, please see http://journals.plos.org/plosone/s/data-availability.

Reviewers' comments:

Reviewer's Responses to Questions

**Comments to the Author**

1. Is the manuscript technically sound, and do the data support the conclusions?

Reviewer #1: Yes

Reviewer #2: Partly

2. Has the statistical analysis been performed appropriately and rigorously? 

Reviewer #1: Yes

Reviewer #2: N/A

3. Have the authors made all data underlying the findings in their manuscript fully available?

Reviewer #1: Yes

Reviewer #2: Yes

4. Is the manuscript presented in an intelligible fashion and written in standard English?

Reviewer #1: Yes

Reviewer #2: No

5. Review Comments to the Author

Reviewer #1: The authors compared the metabolic profiles in cerebrospinal fluid from ME/CFS and GWI and sedentary control after either rest or exercise. The study is concise and sound, but some minor revision should be corrected before further publication.

1. Line 266-272. It is confused that the authors compared the PC aa C42:6 levels between cfs0 and STOPP (GWI post exercise) groups, because multiple factors can contribute to the differences. Though GLM used, I think it is meaningless to compare these two groups.

2. Line 281, line 315-332. The explanation of C5-OH should be careful. The C5-OH, as the metabolite of leucine converted carnitine form is different from the long chain acylcarnitine. The elevation of acyl carnitine can also attributed to increased fatty acid oxidation after exercise (Metabolism. 2002 Apr;51(4):460-4.) The elevation of C5-OH may be the result of activated leucine metabolism. Previous study has showed the positive correlation between short chain acylcarnitine levels and physical performance (DOI: 10.1096/fj.202000493R).

3. The quality of figure 1 should be improved

Reviewer #2: The manuscript entitled “Metabolic biomarkers in cerebrospinal fluid from Gulf War Illness (GWI) and Myalgic Encephalomyelitis / Chronic Fatigue Syndrome (ME/CFS)” by Baraniuk et al submitted to PLOSOne highlighted the Dysfunction in lipid metabolism may distinguish ME/CFS from GWI. Although the manuscript addresses an important question, the manuscript is not written well, lack of coherence, less organized and difficult to follow. I recommend improving the overall presentation for publication.

6. PLOS authors have the option to publish the peer review history of their article (what does this mean?). If published, this will include your full peer review and any attached files.

Reviewer #1: **Yes: **Dezhen Wang

Reviewer #2: No

---

## [Author Response · Author response to Decision Letter 0]

31 Oct 2020

Reviewer #1: The authors compared the metabolic profiles in cerebrospinal fluid from ME/CFS and GWI and sedentary control after either rest or exercise. The study is concise and sound, but some minor revision should be corrected before further publication.

1. Line 266-272. It is confused that the authors compared the PC aa C42:6 levels between cfs0 and STOPP (GWI post exercise) groups, because multiple factors can contribute to the differences. Though GLM used, I think it is meaningless to compare these two groups.

Deleted.

2. Line 281, line 315-332. The explanation of C5-OH should be careful. The C5-OH, as the metabolite of leucine converted carnitine form is different from the long chain acylcarnitine. The elevation of acyl carnitine can also attributed to increased fatty acid oxidation after exercise (Metabolism 2002 Apr;51(4):460-4.). The elevation of C5-OH may be the result of activated leucine metabolism. Previous study has showed the positive correlation between short chain acylcarnitine levels and physical performance (DOI: 10.1096/fj.202000493R).

Thank you for the references regarding “Long- and medium-chain fatty acid oxidation is increased in exercise-trained human skeletal muscle” and “Low intramuscular levels of short-chain acylcarnitines have been found in older females compared to males”.

As a result, we investigated the C5-OH findings from 3 novel perspectives and added limitations and potential pitfalls: 

Line 308 to 363: 

Acylcarnitine C5-OH (C3-DC-M) was higher in postexercise SC than other groups. This suggested (i) no differences between nonexercise groups, (ii) effects of exercise (SC>sc0) and (iii) differences between postexercise SC and GWI groups. …

3. The quality of figure 1 should be improved

Figure 1 has reformatted. The Pearson correlation data for each group have been placed in Supplementary Information as Excel files. The 6 panels are now shown as individual color coded Excel sheets to show correlations for each metabolite.

Thank you for your recommendations.

 

Reviewer #2: The manuscript entitled “Metabolic biomarkers in cerebrospinal fluid from Gulf War Illness (GWI) and Myalgic Encephalomyelitis / Chronic Fatigue Syndrome (ME/CFS)” by Baraniuk et al submitted to PLOSOne highlighted the Dysfunction in lipid metabolism may distinguish ME/CFS from GWI. Although the manuscript addresses an important question, the manuscript is not written well, lack of coherence, less organized and difficult to follow. I recommend improving the overall presentation for publication.

We have had an external review of the paper, and have rewritten substantial portions. These are indicated by the highlighted sections of the revised manuscript. 

The Introduction and Methods are redundant as they describe the 6 subgroups of subjects. We have reinforced the algorithm for assigning group and disease status because it is imperative to emphasize the objective route taken to define characteristics of each subgroup and then to generate new hypotheses to explain the diverse underlying pathological mechanisms.

The results and discussion have been edited as per Reviewer 1. This provides additional discussion of C5-OH, other acylcarnitines and potential dysfunction of metabolomics pathways.

Thank you for your recommendation.

---

## [Decision Letter · Decision Letter 1]

3 Dec 2020

Exercise modifies glutamate and other metabolic biomarkers in cerebrospinal fluid from Gulf War Illness and Myalgic Encephalomyelitis / Chronic Fatigue Syndrome

PONE-D-20-29578R1

Dear Dr. Baraniuk,

We’re pleased to inform you that your manuscript has been judged scientifically suitable for publication and will be formally accepted for publication once it meets all outstanding technical requirements.

Kind regards,

Pankaj K Singh, Ph.D.

Academic Editor

PLOS ONE

Additional Editor Comments (optional):

Reviewers' comments:

Reviewer's Responses to Questions

**Comments to the Author**

1. If the authors have adequately addressed your comments raised in a previous round of review and you feel that this manuscript is now acceptable for publication, you may indicate that here to bypass the “Comments to the Author” section, enter your conflict of interest statement in the “Confidential to Editor” section, and submit your "Accept" recommendation.

Reviewer #1: All comments have been addressed

Reviewer #2: All comments have been addressed

2. Is the manuscript technically sound, and do the data support the conclusions?

Reviewer #1: Yes

Reviewer #2: Partly

3. Has the statistical analysis been performed appropriately and rigorously? 

Reviewer #1: Yes

Reviewer #2: Yes

4. Have the authors made all data underlying the findings in their manuscript fully available?

Reviewer #1: Yes

Reviewer #2: Yes

5. Is the manuscript presented in an intelligible fashion and written in standard English?

Reviewer #1: Yes

Reviewer #2: Yes

6. Review Comments to the Author

Reviewer #1: the authors have adequately addressed my comments raised in a previous round of review and I think this manuscript is now acceptable for publication

Reviewer #2: (No Response)

7. PLOS authors have the option to publish the peer review history of their article (what does this mean?). If published, this will include your full peer review and any attached files.

Reviewer #1: **Yes: **Dezhen Wang

Reviewer #2: No

---

## [Editor Report · Acceptance letter]

23 Dec 2020

PONE-D-20-29578R1 

Exercise modifies glutamate and other metabolic biomarkers in cerebrospinal fluid fromGulf War Illness and Myalgic Encephalomyelitis / Chronic Fatigue Syndrome 

Dear Dr. Baraniuk:

I'm pleased to inform you that your manuscript has been deemed suitable for publication in PLOS ONE. Congratulations! Your manuscript is now with our production department. 

Kind regards, 

on behalf of

Dr. Pankaj K Singh 

Academic Editor

PLOS ONE